# Exploring the Impact of Multimodal Access on Property and Land Economies in Shanghai's Inner Ring Districts: Leveraging Advanced Spatial Analysis Techniques

Wei He [1], Ruqing Zhao [2,*] and Shu Gao [3]

[1] School of Architecture and Allied Art, Guangzhou Academy of Fine Arts, Guangzhou 510006, China; gafahew@gzarts.edu.cn
[2] Department of Geoscience and Natural Resource Management, Faculty of Science, University of Copenhagen, 1870 Copenhagen, Denmark
[3] Texas Department of Transportation, Austin, TX 78744, USA; sgao7@utexas.edu
[*] Correspondence: kzt283@alumni.ku.dk

**Abstract:** This study explores the impact of accessibility on property pricing and land economies by advanced spatial analysis techniques, focusing on Shanghai as a representative metropolis. Despite the impact of metro systems on residential property values, which has been frequently assessed, a research gap exists in understanding this phenomenon in Asian, particularly Chinese, urban contexts. Addressing this gap is crucial for shaping effective urban land use policy and improving the land economy rationally in China and similar settings facing urban challenges. To assess the impact of metro station accessibility on property prices in Shanghai, with extensive rail transit, and to deeply explore the overall impact of land value varieties driven by metro on urban development, we conducted a comprehensive analysis, with discussion about future aspirations for land planning and management along with landscape and facility design, and measures to improve land economy. The procedures involved creating neighborhood centroids to represent accessibility and using the Euclidean distance analysis to determine the shortest paths to metro stations. Our evaluation incorporated a hedonic pricing model, considering variables like neighborhood characteristics, housing attributes, and socio-economic factors. Advanced spatial analysis encompassing Ordinary Least Squares (OLS) regression and XGBoost analysis were employed to explore spatial effects, and Geographically Weighted Regression (GWR) helped examine spatial patterns and address autocorrelation challenges. Results revealed a negative association between distance to metro station and property prices, indicating a non-linear and spatially clustered relationship and heterogeneous spatial pattern. We dissected the non-linear results in detail, which complemented the conclusion in existing research. This study provides valuable insights into the dynamic interplay between metro accessibility and housing market behaviors in a significant Asian urban context, offering targeted suggestions for urban planners and governors to decide on more reasonable land use planning and management strategies, along with landscape and infrastructure design, to promote not only the healthy growth of the real estate market but also the sustainable urban development in China and similar regions.

**Keywords:** land economic; land management; property price; accessibility; hedonic model; spatial analysis techniques; Shanghai

## 1. Introduction

Urban rail transit, encompassing metro and train, stands out as a vital mode of public transportation due to its high speed, efficiency, and safety advantages, garnering greater scholarly attention compared to other investigated transport modes [1,2]. Metro commuting efficiently accommodates more passengers than motor vehicles, mitigating urban traffic congestion caused by sprawl and rapid development. This fosters an efficient lifestyle,

enabling increased social engagement and easier employment access [3]. However, whether the rail transit system has a favorable influence on residential property prices has not yet been drawn to an exact conclusion by prior studies. Comprehending this influence is crucial for the development of land and social economies, as it can inform decisions related to urban land planning and management.

To date, the research on the relationship between rail transit accessibility and real estate value has shown mixed results in the literature, ranging from positive to negligible or negative effects [4]. Gatzlaff and Smith (1993) claimed that the disparity in the results of the empirical studies can be attributed to local factors in each city [5]. Among the studies, North America and Europe have received the most attention. In contrast, limited studies focused on developing countries because of transport investment constraints and the lack of available and reliable data [6]. Many researchers found that metro accessibility had a profitable impact on property value. Bajic (1983) surveyed Toronto and found that the establishment of the subway boosted the average home market value by $2237 [7]. Benjamin and Sirmans (1996) evaluated the impact of the proximity to Buffalo, NY, on residential property values. Houses within a quarter-mile radius received a premium of 2–5% of the median for urban homes [8]. Dorantes, Paez and Vassallo (2011) assessed the impact of proximity to metro stations in Madrid, Spain, showing that metro accessibility had a positive impact on property value and was more pronounced at sale [9]. Mayor et al. (2012) found that in Dublin, a prominent increase showed in prices when the property was near rapid transit DART or light rail Luas [10]. However, some studies found that the effect was not significant. Gatzlaff and Smith (1993) examined the impact of the development of the Miami metro system on the property value near its station location, indicating that the rail system had little effect on residential value [5]. Moreover, Vessali (1996) suggested that while accessibility brought an increment to average residential property value, several complementary factors, such as supportive local land use policies and existing high-density demand development, needed to be considered [11]. Others came to the opposite conclusion. They claimed that proximity to metro stations had a negative impact [12,13].

Discussions based on the Western context may not completely fit Asian cities with high development density and high metro usage rates, where more diverse results may arise. In recent years, some relevant studies have covered prime metropolitan areas in Asia, aiming at providing practical insights for decision-making process of urban land planning and management in the specific context of Asia, to better meet the demands of urban development and contribute to land and socio-economic growth. Bae et al. (2003) investigated the effects of the opening of the new metro line in Seoul on nearby property values and claimed that metro station accessibility had a statistically significant impact on residential prices, consistent with the expected effects observed in previous studies [14]. Chin et al. (2020) studied the same project in Seoul, saying that in some blocks, the setting of the metro line was related to higher increases in apartments, but in other neighborhoods, a reducing trend in price was shown [15]. However, Diao et al. (2016) indicated negative external effects on property values due to railway noise based on research in Singapore [16], while Anantsuksomsri and Tontisirin (2015) studied the Bangkok metropolitan area and found that the closer distance to large traffic stops meant a lower property value [17]. Some studies have been carried out in China. Zheng and Kahn (2008) surveyed Beijing and claimed that public transport infrastructure, including the metro, has been taken into account in property prices [18]. Li (2017) also examined Beijing and concluded that consumers were willing to pay more to use rail transit in congested areas [19]. Yang et al. (2020) documented that metro accessibility affected property prices in a non-linear manner in Shenzhen [20]. Although Shanghai possesses a very well-developed rail transit system, and its dependence on rail transit can be seen to a high extent, research in Shanghai is still quite limited. One of the studies that analyzed the construction of Line 6 in Pudong District showed that the transportation improvement to the CBD employment center brought about an appreciation of property prices in real estate along Line 6 [21]. The

average appreciation was 3.75%, comparable to the data in a previous study in Toronto [7]. It also dissected that the most increases happened in remote suburbs and lower-income communities. However, the study lacks the scale and validity to fully demonstrate how the rail transit system performs in the Shanghai context. There is also research that is available to control the specific characteristics related to real estate to better identify the value of improved accessibility incorporated into property prices.

To fill the knowledge gaps indicated above, we applied a hedonic price model to investigate the impact of metro accessibility on property prices in Urban Shanghai to obtain a broad view and precise conclusion. We aim to offer a profound insight into metro system planning and a reference for the exploitation of real estate and other types of land use to facilitate balanced land and socio-economic development. By applying methods of advanced spatial analysis consisting of Ordinary Least Squares (OLS) regression, XGBoost analysis, and Geographically Weighted Regression (GWR), we approached to evaluate: (1) the relationships between metro accessibility and property prices in Urban Shanghai, and (2) the spatial variation of the influence.

## 2. Materials and Methods

### 2.1. Study Area

Shanghai is widely considered as a city with a single-center structure [22], and is divided into three concentric rings according to the inner and outer rings, including the inner city, expanded inner city and suburbs [23,24]. Our study area is the inner city, popularly known as Urban Shanghai, including districts of Yangpu, Hongkou, Putuo, Changning, Xuhui, Huangpu and Jing'an (see Figure 1). This is the core hinterland of Shanghai, covering an area of 289.59 square kilometers, and is home to 6.88 million people. The broad area and large population establish a solid foundation for investigation because it can provide various samples for the real estate market. As a metropolitan area, Shanghai has perfect facilities, well-developed industry and commerce, and a diverse cultural life, all of which are factors that influence the trend of property prices. The rich construction background and large database enable us to point to more accurate results and make more reasonable suggestions for the future planning of facility construction and property distribution in the city.

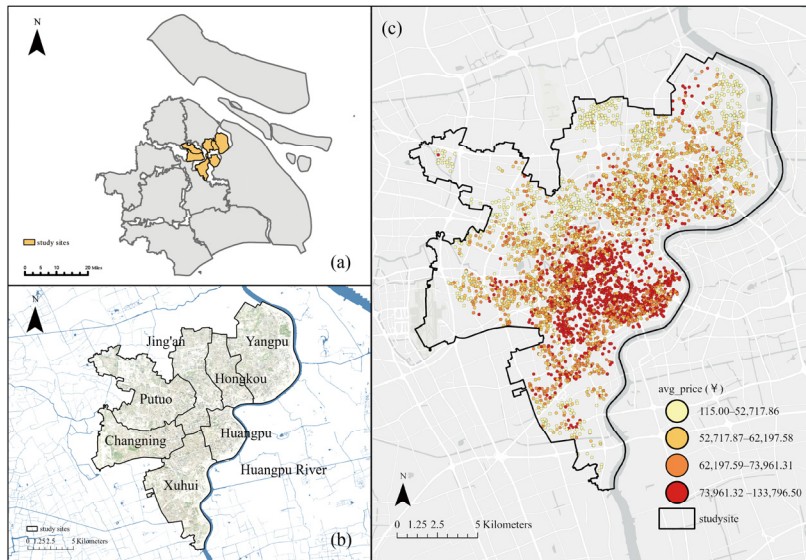

**Figure 1.** Study area. (**a**) Shanghai inner ring area. (**b**) The names and locations of districts within Shanghai Inner Ring and Huangpu River. (**c**) The average property prices.

### 2.2. Data Sources

Table 1 presents a comprehensive overview of the data utilized in this project within the Shanghai Inner Ring region, encompassing administrative boundary data, building

data, property prices, metro station travel flow data, and Shanghai GDP data. All data underwent processing within the WGS 1984 UTM Zone 51N coordinate system. The table provides details on each data source, including the data name, year, resolution, usage, and the corresponding source link.

**Table 1.** Data Sources.

| Data Name | Year | Usage | Data Source |
|---|---|---|---|
| Shanghai GDP | 2019–2020 | Calculating neighborhoods' social-economic attributes | http://www.dsac.cn/DataProduct (accessed on 12 November 2023) |
| Shanghai buildings | 2020 | Extracting building attributes | https://www.amap.com/ (accessed on 12 November 2023) |
| Shanghai AOI—buildings, roads, water systems, subways, administrative divisions | 2020 | Dividing neighborhoods, facilities and boundaries | https://www.amap.com/ (accessed on 12 November 2023) |
| Shanghai property price transaction data | 2019–2020 | Calculating neighborhoods' property prices | https://m.anjuke.com/bj/ (accessed on 12 November 2023) |

*2.3. Framework*

Figure 2 depicts the research flow of this project, in which the chart on the left is the ensemble of the database we needed, and the one on the right is a flow of processing and analyzing data. First, to have an integrative simulation of property prices, the hedonic model proposed by Rosen (1974) was used [25]. Facility data were extracted, and the accessibility data were calculated based on neighborhood units within the boundary of Shanghai's inner ring to build up the hedonic model. Second, a series of correlation analyses including typical linear model Ordinary Least Squares (OLS) and non-linear XGBoost were performed, before which the Pearson correlation test and VIF were conducted to check collinearity. To explore the spatial autocorrelation between variables and the influences of hedonic attributes, we applied Geographically Weighed Regression with an advanced step of checking Moran's I. With these three models, we evaluated comprehensively the relationship between accessibility to metro stations and property values.

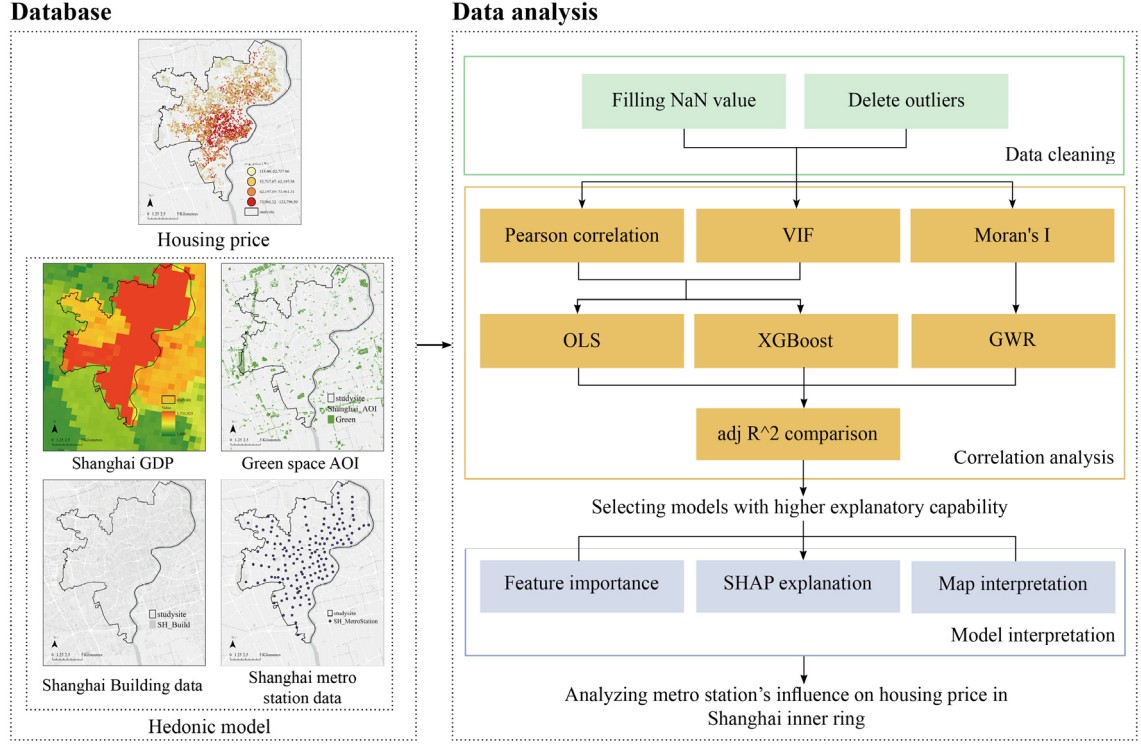

**Figure 2.** Proposed Flowchart of This Research.

*2.4. Hedonic Model*

The hedonic model has been applied in most previous studies to test the acquisition of property values in housing studies. The earliest study, conducted by Dewees (1976), analyzed the relationship between rail travel costs and residential property values [26]. The hedonic model claims that the cost of a specific property can be unveiled through a set of implicit features. A fundamental assumption of the model is that the residential market is determined by a series of choices made by consumers and producers under market conditions [27], and the variations in real estate prices can be explained by the buyer's inclination to invest in various attributes that impact the property's value. The series of characteristics that affect the property price is composed of the housing structure and the external environment. The property price hedonic model applied in this project is expressed as the following Equation (1), which is the most commonly used specification in hedonic property price models:

$$P = \beta_0 + \beta_H H + \beta_N N + \beta_S S + \varepsilon \tag{1}$$

where $P$ is a vector of property prices, $H$ is a matrix of housing structural attributes, $N$ is a matrix of neighborhood attributes, and $S$ is a matrix of socio-economic attributes. $\beta_0$ is the constant term vector, $\beta_H$, $\beta_N$ as well as $\beta_S$ are matrices of the corresponding parameters, and $\varepsilon$ is a vector of error terms. In our Hedonic model, V (property price) was taken as the dependent variable, and 12 implicit variables in total were employed, categorized as I (housing structural attributes including number of floors and number of rooms), E1 (neighborhood attributes composed of accessibility from residential areas to metro stations, schools, grocery stores, etc.), E2 (socio-economic attributes mainly with GDP data), were taken as explanatory variables.

We used data for the Shanghai road network and points of interest (POI). POI includes metro stations, schools, grocery stores, green spaces, restaurants, hospitals, and shopping malls. To aggregate accessibility from residential areas to facilities, centroids for each neighborhood were generated. To calculate the shortest paths, we used the Euclidean distance which can provide a visual representation of accessibility and be easily computed using geographic coordinates fitted in with many geographic information system (GIS) tools and technologies:

$$d_{ij} = \sqrt{\left(x_i - x_j\right)^2 + \left(y_i - y_j\right)^2} \tag{2}$$

where: $x_i$ and $y_i$ = X and Y coordinates of point i with a plane projection. To gain a comprehensive understanding of metro stations, apart from accessibility data, the study delved into flow levels and the number of employees within a 10 min walking distance of each station. These metrics were further categorized into low, medium, and high levels based on their distribution, providing a nuanced perspective on the dynamics of metro station utilization and surrounding employment concentrations.

*2.5. Correlation Analysis*

A set of three models was employed to assess the relation between neighborhoods' shortest distance to metro stations and average property prices: (1) typical linear model Ordinary Least Squares (OLS); (2) non-linear XGBoost which applies a tree-based algorithm for predicting and interpreting data; (3) Geographical Weighed Regression where the spatial autocorrelation between variables is considered.

2.5.1. Ordinary Least Squares (OLS)

The OLS method is the simplest and most applied method among common regression models for analyzing the correlations between two or more variables. The theoretical basis of OLS is that the model coefficients assume that the sample regression model is closest to

the observed value, and the coefficients are constant relative to the location. OLS and the coefficient estimation matrix are represented by Equation (3):

$$y = X\beta + \varepsilon \tag{3}$$

where $y$ is the dependent variable, $X$ is the independent variable, $\varepsilon$ is the deviation while estimating the coefficients.

However, the drawback of this method in spatial modeling is that the value of the dependent variable estimated by it has to serve the whole study area; and, in different parts of the region, the value is estimated the same [28,29].

2.5.2. XGBoost

XGBoost is a tree-based boosting machine learning method which has been recently adopted to explain the non-linear relationship between built environment features and travel. Compared with other traditional models, XGBoost has the advantage of high speed and precision, and it is not influenced by collinearity, which means we can contain all the variables though some of them share high correlation [30,31]. The mathematics of XGBoost can be simplified as follows [32]:

For a given dataset $D = \{(x_i, y_i)\}$, $n$ instances and $m$ features fit a function that can best estimate the response variable $\hat{y}_i$ based on explanatory variables $x_i$. The prediction waterlogging depth ($\hat{y}_i$) after $K$ times iteration of sample $i$ is calculated by (4):

$$\hat{y}_i = \sum_{k=1}^{K} f_k(x_i) = y_i^{(K-1)} + f_k(x_i) \tag{4}$$

where $K$ is the number of iterations and $f_k(x_i)$ is the tree model after k-th iteration.

The objective function of XGBoost is written as (5) and (6):

$$L = \sum_{i=1}^{n} l(y_i, \hat{y}_i) + \sum_{k=1}^{K} \Omega(f_k) \tag{5}$$

$$\Omega(f_k) = \gamma T + \frac{1}{2}\lambda|\omega|^2 \tag{6}$$

where $n$ is the amount of data to improve the k-th tree. $l(y_i, \hat{y}_i)$ is the training loss with target $y_i$ and prediction $\hat{y}_i$. Training loss should be minimized by RMSE. $T$ is the number of leaves and $|\omega|$ is the weight of the leaf $i$. $\gamma$ and $\lambda$ are all hyperparameters.

The K-th base learner function is calculated by summing all weights of leaves (7) and (8):

$$f_k(x_i) = \eta \sum_{I=1}^{T} \omega_{ik} I \tag{7}$$

$$I = \{i|q(x_i) = j\} \tag{8}$$

$\omega$ ranges from 0 to 1, to control the learning rate of iteration to avoid over-fitting. $q(x_i)$ reflects the specific leaf node. $I$ is the sample set of $j$ nodes. To interpret XGBoost, we applied Shapely Additive Explanations (SHAP). SHAP is a Python package for model interpretation. For each explanatory variable, this package returns a SHAP value, which indicates each explanatory variable's influence on the response variable, either intensity or magnitude. The mathematics of SHAP value are as follows (9):

$$g(z') = \theta_0 + \sum_{j=1}^{M} \theta_j z\prime_j \tag{9}$$

where $g$ is each explanatory variable, $M$ is the total amount of explanatory variables and $\theta_j$ is feature $j$'s attribution.

2.5.3. Geographically Weighted Regression (GWR)

Considering the difficulty of simultaneously analyzing all kinds of influence levels and properly dealing with the autocorrelation between different factors, Geographically Weighted Regression (GWR) was adopted to analyze its spatial pattern. This method was first developed by Fotheringham, Charlton, and Brunsdon, professors at British Universities, in 1970 [33]. The first contribution of GWR can be traced back to research in the field of transportation by Du and Mulley (2006, 2007) to compare the impact of hedonic pricing and local modeling methods on land value enhancement [34,35].

GWR is a method for modeling spatial heterogeneity to achieve higher accuracy in analyzing location-affected correlations, which indicates that in each geographical coordinate, the correlation between the dependent and independent variables is different according to the variety of coefficients of the model on the point [33,36]. The coefficients of the model can be estimated at any point in the place. Within this method, observations around points in the place can be used to estimate the model coefficients at each point, while coefficients with closer observations are given greater weight. It is worth noting that each coefficient of GWR has a value with a sign. The GWR linear multivariate regression model is represented by Equation (10):

$$y_i(u) = \beta_{0i}(u) + \beta_{1i}(u)x_{1i}\beta_{2i}(u)x_{2i} + \beta_{mi}(u)x_{mi} \tag{10}$$

where $y$ is the vector of the observed value, $x$ is the matrix of the independent variables and $\beta$ is the estimated coefficient vector.

Before delving into the correlation analysis, a pre-processing process was initiated. This involved subjecting the dataset to scrutiny through both Pearson correlation and variance inflation factor (VIF) assessments, aimed at forestalling collinearity among explanatory variables. The meticulous identification and mitigation of collinearity are essential steps in preserving the integrity and interpretability of the subsequent analyses.

Furthermore, to enhance the robustness of the dataset, a stringent approach was taken to address outliers. Employing z-scores with a threshold set at 3, outliers were systematically identified and subsequently removed. This meticulous outlier removal process contributes to the reliability of the subsequent statistical analyses by mitigating the undue influence of extreme values. Additionally, addressing missing data is another crucial facet of data preparation. The strategy employed involved filling in missing values with the mean of each respective column, ensuring a balanced and representative dataset.

**3. Result**

*3.1. Hedonic Model*

At the core of this investigation lies the dependent variable, the average property price around which an intricate web of explanatory variables revolves. The selection of these variables was guided by the hedonic model, categorizing them into three overarching themes: structural attributes, neighborhood attributes, and socio-economic attributes (see Table 2).

To ensure a comprehensive analysis, data pertaining to the centroids of all neighborhoods, totaling 4627, was meticulously gathered for examination, with structural housing attributes, including the number of floors and number of rooms, as explanatory variables. From Figure 3, both variables were relatively randomly distributed.

**Table 2.** Statistics of explanatory variables.

| | Variable Name | Description | Mean | Std. Dev. | Data Type | Unit |
|---|---|---|---|---|---|---|
| Structural attributes | Floor | Average number of floors per neighborhood | 4 | 2 | Continuous | |
| | Number of rooms | Average number of rooms per neighborhood | 2 | 1 | Continuous | |
| Neighborhood attributes | Distance to restaurants | Shortest Euclidean distance between each neighborhood and restaurants | 719.89 | 411.94 | Continuous | meter |
| | Distance to groceries | Shortest Euclidean distance between each neighborhood and groceries | 707.32 | 408.88 | Continuous | meter |
| | Distance to schools | Shortest Euclidean distance between each neighborhood and schools | 258.97 | 164.36 | Continuous | meter |
| | Distance to hospitals | Shortest Euclidean distance between each neighborhood and hospitals | 514.03 | 290.34 | Continuous | meter |
| | Distance to greenery | Shortest Euclidean distance between each neighborhood and green space | 452.62 | 254.16 | Continuous | meter |
| | Distance to shopping centers | Shortest Euclidean distance between each neighborhood and shopping centers | 1998.49 | 1020.38 | Continuous | meter |
| | Flow level | Average flow level on the day of metro stations | 3 | 1 | Ordinal | |
| | Distance to metro stations | Shortest Euclidean distance between each neighborhood and metro stations | 575.91 | 318.43 | Continuous | meter |
| | Number of employments | Number of employment points within walking distance of each metro station | 48.16 | 32.49 | Continuous | |
| Social-economic attributes | GDP | Average GDP per neighborhood | 598,218.69 | 497,457.74 | Continuous | ¥ |

For neighborhood attributes, we calculated the Euclidean distances between facilities (restaurants, grocery stores, schools, hospitals, greenery space, shopping centers, and metro stations) and neighborhoods as accessibility input. Besides the accessibility data to obtain details of metro stations, passenger flow levels and the number of employment centers were calculated, as shown in Figure 4. From the maps, metro stations located in the center of Shanghai's Inner Ring region have higher passenger flow levels and a larger number of available employment centers. Neighborhoods near these metro stations have relatively shorter Euclidean distances to the stations.

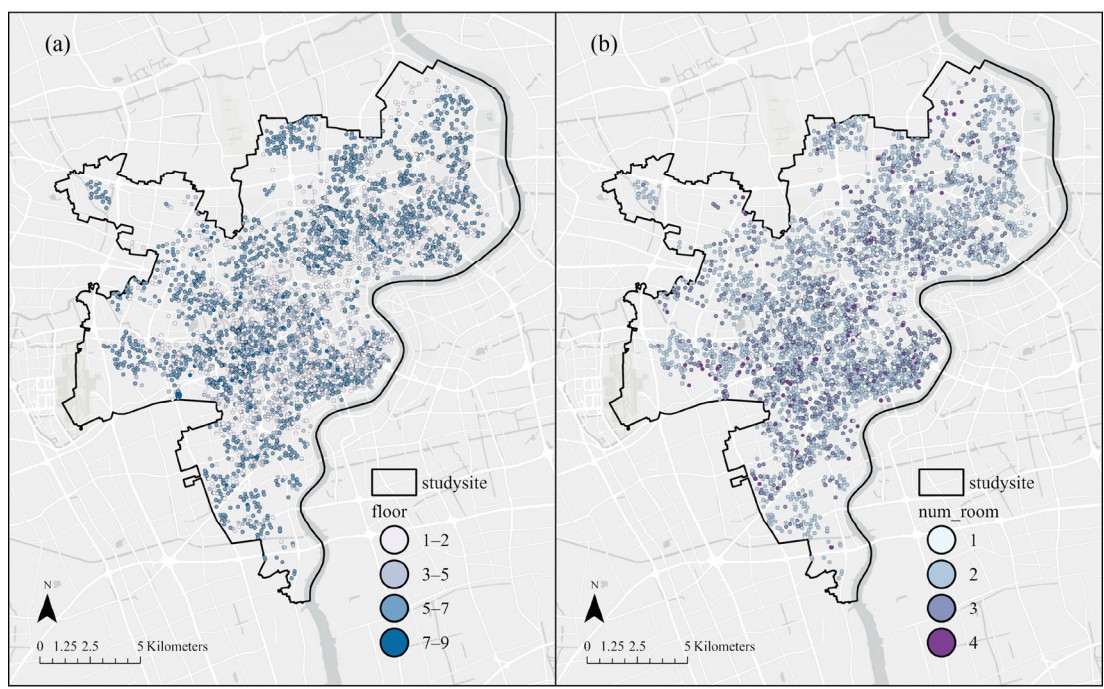

**Figure 3.** (**a**) Illustration of number of floors per neighborhood. (**b**) Illustration of number of rooms per neighborhood.

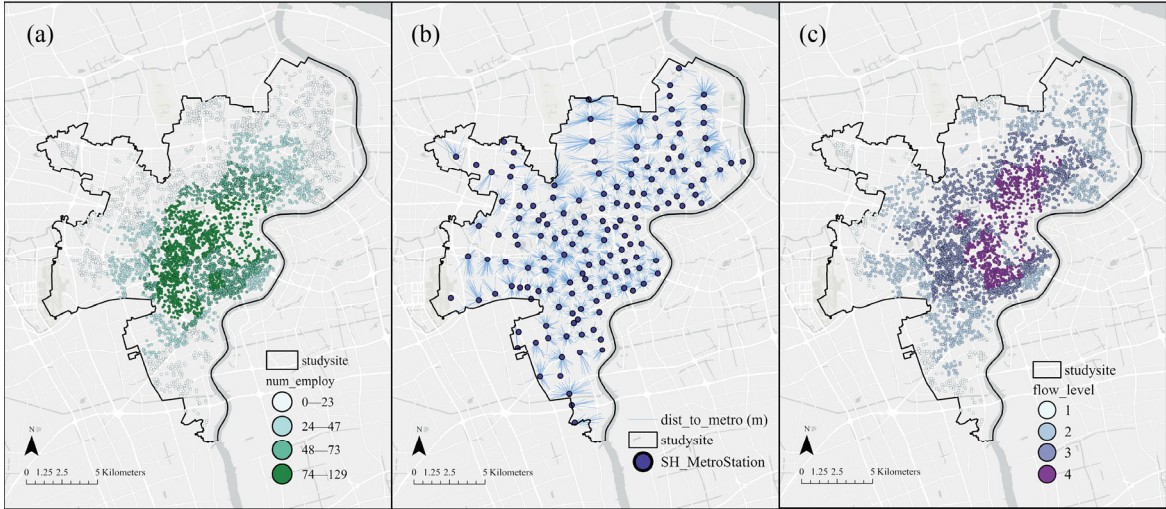

**Figure 4.** (**a**) Number of available employment centers per metro station. (**b**) Example of shortest distance to metro stations. (**c**) Passenger flow level per metro station.

In addition to the data related to accessibility and housing construction, to enhance the model accuracy, we collected GDP data of each neighborhood as economic factors that have the potential to influence the result. As shown in Figure 5, central areas along the Huangpu River have the highest average GDP, and neighborhoods in the west are in relatively low GDP areas within Shanghai's Inner Ring region.

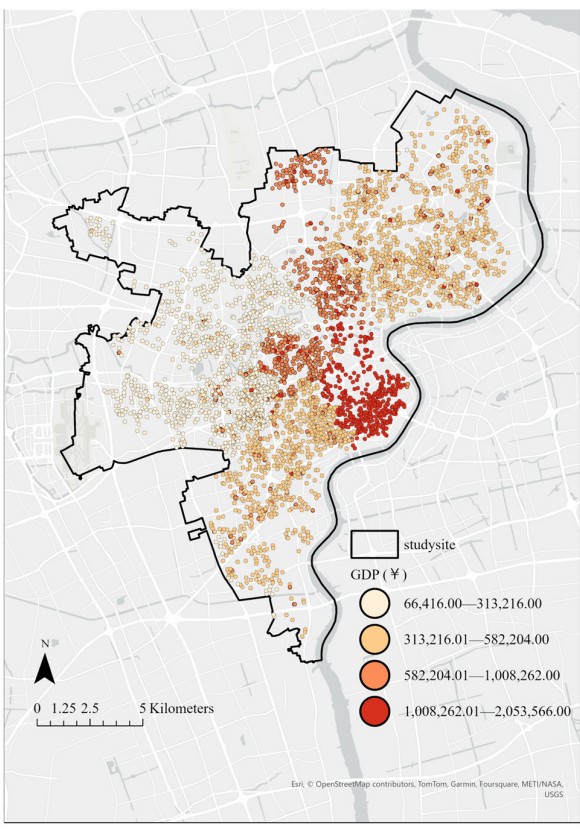

**Figure 5.** Illustration of GDP per neighborhood.

### 3.2. Correlation Analysis

Through data-cleaning, a total of 4170 neighborhoods emerged as the refined dataset, ready for a more nuanced and accurate exploration of the relationships between explanatory variables and average property prices. Before doing correlation analysis, the Pearson correlation test and VIF were conducted to check collinearity. Pearson results and VIF were all below 0.7, indicating that no collinearity existed in the input dataset.

Moran's I was checked before implementing the Geographically Weighted Regression model (GWR). The result of 0.50 with a *p*-value less than 0.05 indicated that the residuals were significantly clustered. The involvement of GWR could effectively solve this problem and better explain the relation between our target variables.

#### 3.2.1. OLS & XGBoost Model

Table 3 serves as a comprehensive repository delineating the myriad explanatory variables employed in this research endeavor, accompanied by their corresponding metrics. This tabular exposition offers a profound insight into the dynamics of average property prices within the analytical frameworks of both Ordinary Least Squares (OLS) and XG-Boost models.

**Table 3.** Correlation Analysis Results.

| Variables/Models | OLS | | XGBoost |
|---|---|---|---|
| | Coefficient | $p > |t|$ | Feature Importance |
| Intercept | 0.425 | 0.0 *** | / |
| Structural attributes | | | |
| floor | −0.032 | 0.0 *** | 0.090 |
| Num_room | 0.085 | 0.0 *** | 0.060 |
| Neighborhood attributes | | | |
| dist_rest | −0.067 | 0.0 *** | 0.085 |
| dist_grocery | 0.045 | 0.0 *** | 0.045 |
| dist_school | −0.027 | 0.009 ** | 0.035 |
| dist_hospital | −0.017 | 0.08 ** | 0.030 |
| dist_greenery | −0.014 | 0.138 | 0.022 |
| dist_shopping | −0.037 | 0.0 *** | 0.112 * |
| Flow_level | −0.0001 | 0.989 | 0.037 |
| dist_metro | −0.053 | 0.0 *** | 0.053 |
| num_employ | 0.183 | 0.0 *** | 0.307 *** |
| Social-economic attributes | | | |
| GDP | 0.099 | 0.0 *** | 0.124 ** |
| $R^2$ (adj.) | 0.242 | | 0.520 |

Note: For OLS, significance level = 0.01: ***, 0.05: **, 0.1: *; For XGBoost, significance levels = highest feature importance: ***, 2nd highest feature importance: **, 3rd highest feature importance: *.

In dissecting the results derived from the OLS analysis, a discernible and statistically significant relationship was unearthed for most variables associated with average property prices. An intriguing pattern emerged as variables representing the shortest distances, apart from proximity to groceries, predominantly manifested negative impacts on property prices. Noteworthy among the metro station features was the revelation that a number of employees were within convenient walking distance.

Transitioning to the XGBoost model (see Figure 6), a distinct hierarchy of influential features emerged. Foremost among them was the number of employees in the vicinity of metro stations, followed closely by local GDP levels, and, subsequently, the distance to shopping centers. Corresponding with the OLS findings, most variables regarding the shortest distances showcased a positive influence on average property prices in the XGBoost model. However, a salient departure from this trend was observed in the case of the shortest distance to grocery stores.

A pivotal divergence between the two models lies in the higher R-squared value associated with the XGBoost results. When employing the XGBoost model, the elevated R-squared of 0.520 signifies a more nuanced and non-linear relationship between explanatory variables and average property prices, compared to the R-squared of 0.242 derived from OLS model. This underscores the superior capability of XGBoost in capturing intricate and complex relationships within the dataset, which is in stark contrast to the inherently linear nature of the OLS model. The XGBoost model's capacity to discern subtle nuances and non-linear patterns signifies a substantial advancement in predictive modeling, which can better capture the intricate association between factors and prices.

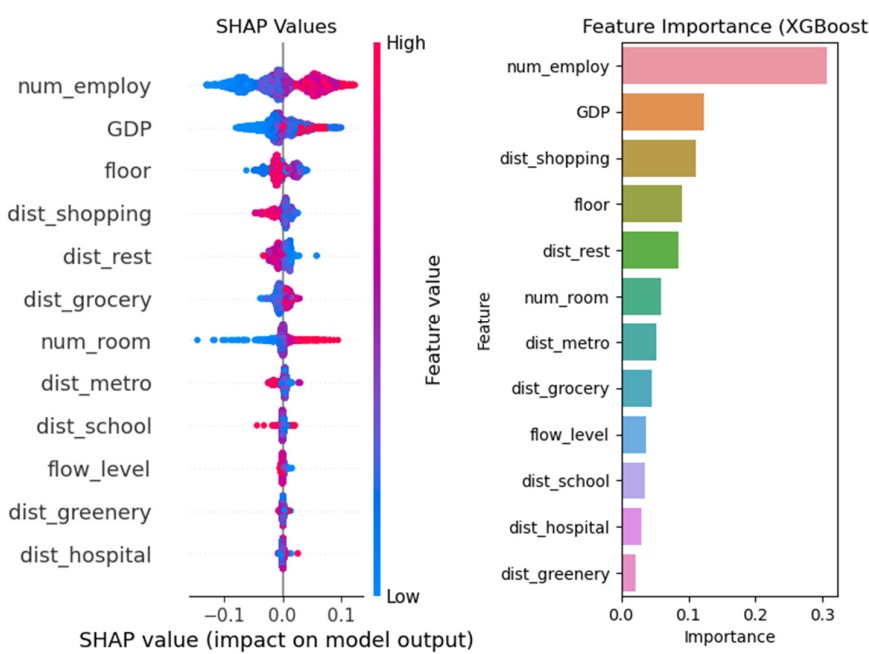

**Figure 6.** Correlation Matrix of explanatory variables.

### 3.2.2. GWR Analysis

After detecting the non-linear statistics, a Geographically Weighted Regression (GWR) model was utilized to better examine the correlation between the proximity of neighborhoods to metro stations and their average property prices. The model yielded an adjusted R-squared of 0.623, indicating a superior explanatory power compared to other models, particularly when accounting for spatial disparities, which evidently revealed the non-linear and spatially clustered relationship and heterogeneous spatial pattern.

In Figure 7, the coefficients for the shortest distance to metro stations in each neighborhood are depicted, revealing evident spatial heterogeneity. Notably, neighborhoods in the central areas along the Huangpu River, such as the Jing'an district and the Huangpu district, exhibited positive coefficients, suggesting that, as the distance to metro stations increased, average property prices also rose. Spatial clusters were demonstrated. Surprisingly, these areas were usually surrounded by neighborhoods where closer proximity to metro stations was associated with higher property prices. This trend represented a reduction in distance to metro stations, corresponding to a significant increase in property prices. As one moves farther away from the core area, the impact of the distance to metro stations on property prices gradually stabilizes.

As shown in Figure 8, according to the coefficients of the distance to the metro in the GWR analysis outcome, we divided the data into five categories. Then, we analyzed the average values of the main factors that influence property prices the most on the basis of the XGBoost model. Among the variables with negative interpretation, the average distance to the metro is quite stable, which is about 505 to 550 m, but varies in the positive interpretation from around 440 to 550 m. This means that to affect real estate value, various factors are inseparably interconnected, such as the number of employees, the distance to a shopping center and the GDP, each of which has a great effect on property prices.

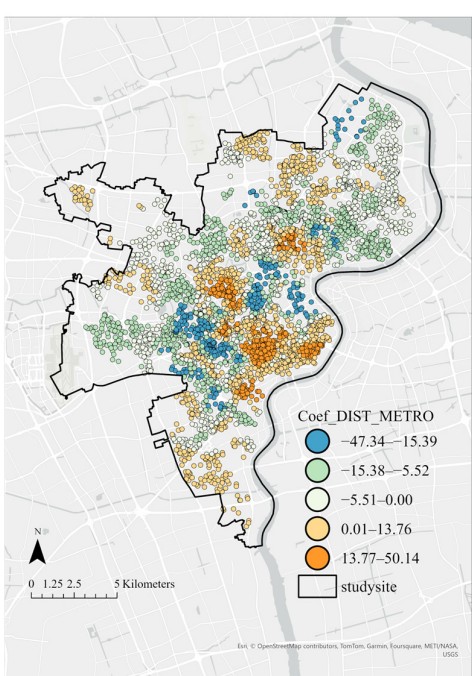

**Figure 7.** GWR regression coefficients of shortest distance to metro stations.

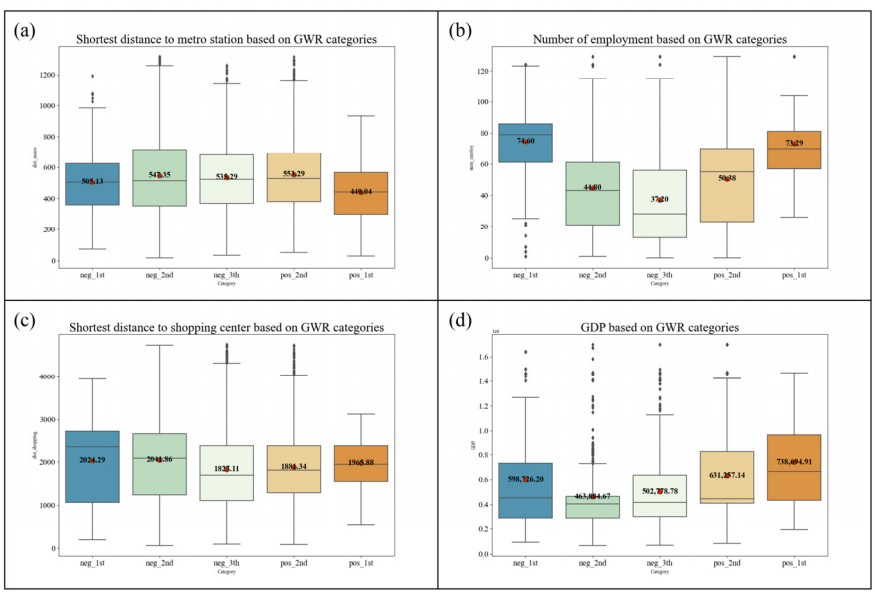

**Figure 8.** Distribution of coefficients average values of the main factors. (**a**) Shortest distance to metro station based on GWR categories. (**b**) Number of employment based on GWR categories. (**c**) Shortest distance to shopping center based on GWR categories. (**d**) GDP based on GWR categories.

## 4. Discussion

### 4.1. Overall Impact of Metro Accessibility on Property Prices: Non-Linear and Spatially Heterogeneous

Our result showed that the accessibility to metro stations was positively related to property prices, which was consistent with many previous studies [7,9,10,26]. Meanwhile, the relationship between the variables displayed a non-linear pattern and showed a spatial aggregation. The spatial autocorrelation had a certain impact on the property price. In the central areas along the Huangpu River, such as the Jing'an district and the Hongkou district, average property prices rose as the distance to metro stations increased. Conversely, in the neighborhoods surrounding the central area, closer proximity to metro stations was associated with higher property prices. On the other hand, in the districts farther away

from the core area, the impact of distance to the metro station on property prices gradually stabilized. Similarly, Bowes and Ihlanfeldt (2001) indicated in their study that the impact of railway stations depended on their distance from the central business district [1]. We also suggested the significance of service facilities in residential areas, as our findings highlighted that accessibility to amenities like restaurants, shopping malls, and schools had a positive impact on property values. A study conducted in Seoul by Bae, Jun and Park (2003) found consistent results, indicating that individuals tended to reside in sub-center areas characterized by a higher concentration of recreational and commercial services [14].

There may be several reasons to explain the results of our study. Proximity to metro services has the potential to improve nearby property values, but metro accessibility alone may be insufficient to bring about significant changes. Other factors, such as economic stimulus, land use policy, and development subsidies should be taken into account when evaluating the influences of metro services on property prices [5]. Accessibility to metro stations also has some adverse effects on nearby communities. Therefore, the comprehensive impacts may be mixed, and the trend in property prices ultimately depends on which factor dominates.

In the central area along the Huangpu River, such as the Jing'an district and the Huangpu district, the city density is relatively higher than that of the surrounding neighborhoods. The bustling business and complex personnel, leading to noise, increased the crime rate and visual intrusion [37], directly a reducing living comfort. When purchasing properties, the downsides considered outweighed the advantages of convenience. As mentioned before (see Figure 5), central areas along the Huangpu River have the highest average GDP and there is an office building cluster in this district, where commuters living nearby may choose to walk and ride bikes (shared or private) to avoid the metro crowds during rush hours. However, in the areas surrounding the core, which is a little farther, traffic is often very congested and blocked, and parking facilities are in short supply. To avoid these two annoying troubles, people may take the metro as a more relaxing option. In a study conducted by Li (2017), it was found that the intersection of multiple metro lines, as well as the 30 min metro to work, results in an additional premium for property prices [19]. Additionally, the superior location of the central area along the Huangpu River, with a view of the river and the convenience of refined facilities, leads to almost the highest property price in the whole of Shanghai. People living in the central area are more likely to belong to the high-income group, who are more inclined to choose more convenient and comfortable private trip modes. Conversely, low-income groups are not affluent enough to afford cars and taxis, and hence prefer public transport like the metro, and always tend to walk or cycle to metro stations. Prior studies have also found that increased metro accessibility did not help property price growth in affluent communities [1,38], but Nelson (1992) claimed this improvement will accelerate the capitalization of low-income communities [39]. He tested the impact of the Atlanta Metropolitan Regional Rapid Transit Authority (MARTA) in an area of DeKalb County and found that the proximity to the metro is positively correlated with the property value in lower-income neighborhoods, with the opposite occurring in high-income areas. People in high-income areas (central areas) were less dependent on the metro, and even public transportation, than people in lower-income areas. Similarly, He (2020) proved that the property value premium due to rail accessibility was more significant in the suburbs than in urban areas in Hong Kong [40].

Besides, according to previous research, metro proximity and other related attributes have a different impact on property prices and rental prices [41]. Due to the expensive living costs in the central core, more people who reside here simply choose to rent rather than buy. A considerable amount of university students or new graduates may dwell in the central area and choose off-campus housing for rent [42]. The monetary effect of improved metro accessibility may somehow be reflected in rent rather than the selling price [43]. Another possible reason is that the strengths of improving accessibility may lie in accelerating the sale of homes rather than increasing their value [35].

### 4.2. Limitations of Our Current Study

We employed three models to gain convincing conclusions and explained the non-stationarity between metro accessibility and property value, i.e., the spatially existing autocorrelation. Although many previous studies have discussed the correlation between property prices and metro (rail transit) accessibility, most were concentrated in mature Western economies, few in the context of developing countries, Asia, and China. Since urban densities and population structures vary from country to country, we cannot fully apply conclusions and suggestions in urban planning from Western country research. Meanwhile, current research on the impact of the metro on the surrounding property prices seems to focus on a certain rail line or discuss it in general (including intercity high-speed rail, etc.). Therefore, we sought to fill these vacancies. The results of this study can provide some ideas for the comprehensive evaluation of metro system construction, including the manifestation of construction and operation costs in external real estate benefits. During metro line planning, the station location can be reasonably arranged to balance the discrepancy of property prices between different living areas of the city and make the urban development of Shanghai more all-round. Exploring lessons from Shanghai as a representative metropolis can offer meaningful insights not only for Chinese cities but also for other fast-growing urban areas across the globe.

Nevertheless, there are several limitations. Since the property transaction data we collected were based on records from different years, certain spatial and temporal variation factors may not be considered in the model. Moreover, the diversification of property prices in the economic environment, as well as market adjustments, might potentially bias our research results. To obtain more precise results, spatio-temporal consistency and longitudinal studies should be incorporated. We have applied a series of variables about the external environment and housing structure in our study to help analyze the impact of metro accessibility on property prices. Whereas in the housing structure, we simply encompassed the number of floors and the number of rooms, attributes like building typology, construction standards, structural technical condition and facility equipment are also significant to real estate values. In addition, the area and age of the properties that would contribute to the fluctuations in prices were neglected, especially in the context of multiple administrative regions, where development initiated and peaked in different periods in each area, greatly affecting the values of the real estate [44]. Properties that were built in different eras were equipped with varying floor area ratios and green space ratios in the residential region, which can also affect real estate prices. Moreover, there may be some other potential covariates influencing the values such as the level of rail service, network connectivity, service coverage, and station facilities [4]. In addition, as mentioned above, many commuters choose to ride from home to the metro station. With the development of the sharing economy, shared bikes may provide them with more convenience, which may also increase the influence of metro accessibility on property prices [45]. Moreover, when defining accessibility variables, we applied the Euclidean distance which simply calculated the straight distance between starting and ending points. The simple assessment of accessibility did not consider the practical state of the street and walking path networks, so it is insufficient to reflect the fact of travel costs or barriers. For further studies, accessibility measures with higher robustness and diversity are suggested.

### 4.3. Prospects for Future Study

Since we have discovered that spatial autocorrelation and heterogeneity can be elements that affected property values, in the future, the related factors can be fixed while applying a statistic model. For instance, as the outcomes showed, the number of employees in the vicinity of metro stations and local GDP levels are two important factors affecting the non-linear. The research in the next phase will be expected to involve or fix more socio-economic factors comprising a matrix of the proportion of different race or ethnic groups of people, the median household income and unemployment rate of the census tract [27]. Moreover, apart from the existing characteristics popularly used, natural

amenities, historical amenities and modern amenities can be supplemented. Among them, natural amenities are always linked with forest coverage (NDVI), Slope, Index of NOX as well as CO.

We hope that the outcomes of our research can provide a reference in the field of property and rail transit, even in a broader scope of the whole urban land use planning and management, as well as land and socio-economic development. In metropolia in developing countries such as Shanghai, the optimization of urban planning and management, including the reasonable land use, transport system and amenity setup, along with emphasis on improving landscape design, is for the sake of promoting local economic growth and improving the quality of residential life. To achieve these goals, attracting real estate development investment is an effective approach [46–48]. In Shanghai, the current urban center is over-concentrated [49]. The government needs to ensure more fairly accessible facility services, address the issue of affordable housing, disperse the traffic flow, and balance the regional development by establishing multiple employment centers and more considerate land use planning and land management. Though we have reached the conclusion of a positive relationship between metro accessibility and real estate values in general, complying with the non-linear pattern, there is potential to explore more deeply, such as the distance thresholds where metro accessibility has an impact on property values. This can inform the land use decision-making of the transition zone between properties and metro stations. In the future, when investigating and evaluating the construction distribution and investment effect of rail transit systems or other infrastructure, we should consider various aspects, not sticking to the results reflected in the real estate price. The improvement of metro accessibility not only has direct influence on property prices, but also a profound impact on urban land planning, design and management. Optimizing the layout of the metro network needs to take into account future urban development orientation to apply the optimal solution of land use, while the balance between the supply and the demand of land should also be considered to avoid the unstable land economic situation. Objectively, the high-efficient allocation of land, along with perpetual land management, the popularization of high-quality public transport systems and the improvement of landscape and facility services, are of great significance to urban development, leading to the agglomeration of high-level employment opportunities, growth of land economy, and promotion of the overall development of cities [50]. Apart from emphasizing accessibility to metro stations, future research can also incorporate additional accessibility to amenities such as parks, grocery stores, and schools [51], while also accounting for negative forces brought on by the proximity, such as vacant lands, crime rates, congestion, air contamination and noise pollution [1,52]. Not only the site selection of the metro stations but also the surrounding land planning and its management, as well as maintenance, are important factors affecting the property price. The measured planning of the land use around both metro stations and real estate, such as commercial, education and public facilities, can have a valid impact on land value and the property price level of the whole area.

## 5. Conclusions

The metro has great potential to enhance the value of residential property by improving convenience and saving transportation budgets. However, whether the metro system has a favorable effect on residential property prices has not reached a unified conclusion through prior studies. Furthermore, Shanghai, as a typical example of a flourishing metropolis in Asia, with a highly dependent and well-developed metro system, has great research value, but few studies based on it have been conducted. To bridge gaps in knowledge, our objective is to gain a comprehensive understanding of the relationship between property prices and metro accessibility in Urban Shanghai and to further explore the spatial pattern of this correlation, which is expected to offer insight into urban land use planning along with land management, and suitable landscape and facility design.

We adopted the Euclidean distance between the metro station and the property to measure metro accessibility. Then, we applied the hedonic model proposed by Rosen

(1974) [25], using the Ordinary Least Squares (OLS) model and the XGBoost model, which is a relatively new analytical method, to identify the statistical relationship between metro accessibility and property prices. Considering the difficulty in simultaneously analyzing the various levels of influence and correctly handling the autocorrelation between different factors, the spatial pattern was analyzed by Geographically Weighted Regression (GWR).

Overall, our OLS and XGBoost results indicated that the distance to the metro station was negatively associated with property prices. In other words, the value of land around metro stations can be raised with the improvement of metro accessibility. This provides great economic potential for land development and attracts investors and developers to participate in the active development of urban land economy. Specifically, XGBoost showed a higher r-square, indicating a non-linear association between the explanatory variables and property prices. The GWR model examined the heterogeneous spatial pattern between the proximity of properties and metro stations and their average property prices. The outcomes of the model indicated that communities in the central area, such as the Jing'an and Hongkou districts, exhibited positive coefficients, suggesting that average property prices rose as the distance to the metro station increased. Conversely, in the surrounding neighborhoods, closer proximity to metro stations meant higher property prices. In addition, our results also indicated that the accessibility variable performance had mainly positive effects, except for groceries, emphasizing the importance of service facilities around residential areas in the formation of property prices. We recognized the complex relationship between property prices and metro accessibility, and provided a certain reference for the urban planning of Shanghai to promote the land and social economies. In the future, more in-depth research is needed to fully understand this correlation and to provide more specific guidance for future urban development regarding land use planning and management with the consideration of land economic growth.

**Author Contributions:** Conceptualization, W.H. and R.Z.; methodology, R.Z. and S.G.; data collection, W.H.; data curation, W.H.; formal analysis, R.Z. and W.H.; visualization, R.Z. and S.G.; project administration, W.H.; supervision, W.H.; software, R.Z. and S.G.; validation, W.H.; writing—original draft, W.H. and R.Z.; writing—review and editing, W.H., R.Z. and S.G. All authors have read and agreed to the published version of the manuscript.

**Funding:** This research received no external funding.

**Data Availability Statement:** The raw data supporting the conclusions of this article will be made available by the authors on request.

**Conflicts of Interest:** The authors have no relevant financial or non-financial competing interests.

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
