# Peer review of "Exploring the Impact of Multimodal Access on Property and Land Economies in Shanghai’s Inner Ring Districts: Leveraging Advanced Spatial Analysis Techniques"

_land, doi:10.3390/land13030311_

Round 1
Reviewer 1 Report
Comments and Suggestions for Authors
The topic of the article is very relevant to the rapidly developing cities of Asia, and is based on a large naturalistic study with data covering 4,627 neighborhoods. A clearly drawn up research plan has been used to evaluate the data using three methods, the results of which complement each other. It can be seen that the purposefully named attractions are relevant for residents, starting with the number of jobs, shopping centers, schools, etc. their availability. The relationship between real estate prices and metro location is very important and is not fully analyzed in the article, because the central districts along the Huangpu River, such as Jing'an District and Huangpu District, where the average real estate prices increased as the distance to the metro stations increased. increased. Conversely, in neighborhoods around the central area, being closer to metro stations was associated with higher property prices. Real estate prices stabilized further away from metro stations. The following questions arise:
When do real estate prices rise near metro stations and what does it depend on?
At what distance from metro stations is the effect felt on property prices?
Which valuation method used most clearly and accurately shows the effect of the location of metro stations on the price of real estate?
It is very valuable that the article indicates the limitations of the current study and the prospects for future studies.
Regarding the limitations of the study, it is partially indicated which variables can be added to the future study to reduce the spatial and temporal dispersion of the data used. The article explains the general trends between real estate prices and the location of metro stations, but it would be appropriate to examine in more detail the location of metro stations in space and their influence on real estate.
Reviewer 2 Report
Comments and Suggestions for Authors
The paper presented here contains proof of the well-known relationship that the closer you get to a metro station, the more property prices rise. After reading most of the text, I thought the paper was about simple conclusions. Nevertheless, the Discussion of the results creates significant added value. In my opinion, the abstract must emphasize the uniqueness of the results. All the more so because, after reading the whole, the Authors have come to a very unusual conclusion, and they present only this basic relationship.
In my opinion, the weakness of the model is the matrix of housing structural attributes. In lines 273-274 the Authors wrote: "Housing structural attributes included the number of floors and number of rooms as explanatory variables." This is the biggest weakness of the text. There has been a simplification of the model to two simple attributes and probably easy to identify. In practice, real estate prices depend heavily on the housing standard and technical condition of the building in which they are located. This has been completely ignored. In my opinion, it should be made clear in the text in which area the authors made simplifications.
Moreover, It is necessary to stress more in abstract the most important result: "non-linear pattern and showed a spatial aggregation" - line 364.
Besides, You wrote: "In the central areas along the Huangpu River, such as Jing’an district and Hongkou district, average property prices rose as the distance to metro stations increased." lines 365-366. Please make it deeper, why?
In my opinion, the greatest added value of this article is the demonstration of the non-linearity of the relationship between housing prices and access to metro stations. What I miss is the explanation of this non-linearity. In my opinion, this article will definitely gain more value with the addition of an explanation. In my opinion, this conclusion is worth highlighting. It is worth considering whether to develop this problem in future articles.
Comments on the Quality of English Language
The paper needs minor corrections.
Reviewer 3 Report
Comments and Suggestions for Authors
Dear authors,
you have written an interesting article according to my opinion and to my field of interest.
The manuscript falls within the scope of the subject matter of the journal and meets its requirements, addressing the important problem of property valuation using advanced spatial analyses. The structure of the article is appropriate and well-ordered, despite the technical nature, the language is understandable, and the authors explain the issues discussed sufficiently. The manuscript describes applied research which has practical value, and the results and methods used are clearly presented. I propose to accept the manuscript for publication after minor revision: Figure 2 needs further clarification. The content of the database is not clear, and it appears that the representation of metro stations is missing. I suggest that this be corrected, and the image be more thoroughly explained in the paper. The text in the legend on Figures 3, 4 and all the other Figures containing thematic maps appears to be too small and difficult to read.
